# Posterior Reversible Encephalopathy Syndrome Following Chemotherapy and Immune Checkpoint Inhibitor Combination in a Patient with Small-Cell Lung Cancer

**DOI:** 10.3390/diagnostics12061369

**Published:** 2022-06-02

**Authors:** Cécile Evin, Nathalie Lassau, Corinne Balleyguier, Tarek Assi, Samy Ammari

**Affiliations:** 1Department of Imaging, Gustave Roussy, Université Paris-Saclay, 94805 Villejuif, France; cecile.evin@aphp.fr (C.E.); nathalie.lassau@gustaveroussy.fr (N.L.); corinne.balleyguier@gustaveroussy.fr (C.B.); 2Biomaps, UMR1281 INSERM, CEA, CNRS, Université Paris-Saclay, 94805 Villejuif, France; 3Department of Oncology, Gustave Roussy, Université Paris-Saclay, 94805 Villejuif, France; tarek.assi@gustaveroussy.fr

**Keywords:** PRES, chemotherapy, immunotherapy, status epilepticus, diaschisis

## Abstract

Posterior reversible encephalopathy syndrome (PRES) is a rare neurological complication that occurs following a sudden blood pressure increase. We report the case of a 64-year-old patient presenting PRES several hours after the administration of a combination of chemotherapy and a checkpoint inhibitor (carboplatin-etoposide-atezolizumab) for small-cell lung cancer. He presented consciousness disorders associated with partial epileptic seizure secondarily generalized. His arterial blood pressure was elevated and brain imaging showed multiple bilateral subcortical parietal, temporal, occipital and cerebellar T2 high signals, predominantly in the posterior region. There were no abnormal T1 signals nor bleeding but a left apparent diffusion coefficient restriction was noted. On arterial spin labelling perfusion sequences, there was an increased perfusion within the left temporo-parieto-occipital, left thalamic and right cerebellar regions. Finally, the neurological symptoms completely regressed after several days of optimal antihypertensive and antiepileptic treatment. The clinical context and radiological features, as well as the progressive resolution of the neurological symptoms, were all in favor of PRES. PRES can occur after the administration of chemotherapy and/or immunotherapy. Prompt diagnosis is crucial through a spectrum of suspicious clinical and radiological characteristics that must be rapidly recognized to quickly anticipate the optimal therapeutic strategy and avoid unnecessary complications.

## 1. Introduction

Posterior reversible encephalopathy syndrome (PRES) is a rare neurological phenomenon that occurs following a sudden blood pressure increase in chronically hypertensive patients [1]. It is characterized by non-specific neurological symptoms such as headache, confusion, seizure, loss of vision or coma [2,3,4]. The proper diagnosis is based on the clinical context and the radiological features, but also on the evolution of the symptoms with time. Chemotherapy-induced PRES was previously reported, most particularly with platinum agents, while only a handful of case reports were published with immune checkpoint inhibitors (ICIs) [5,6,7,8,9,10,11]. Rare immune-related neurological events were found to be associated with the novel ICIs including encephalitis, aseptic meningitis, myelitis and others. In this paper, we report the case of a patient presenting with PRES, only several hours after the administration of the combination of chemotherapy and a checkpoint inhibitor (atezolizumab) for metastatic small-cell lung cancer.

## 2. Case Report

A 64-year-old man was hospitalized in the thoracic oncology department for the initial management of an extensive stage small-cell lung cancer (SCLC) (T1N2M0). He was a heavy smoker (weaned 2 years ago) with several comorbidities including hypertension, ischemic heart disease and diabetes. He was also diagnosed with locally advanced laryngeal squamous cell carcinoma which was treated with a combination of radiation therapy and chemotherapy followed by radical surgery.

The first cycle of carboplatin (area under the curve (AUC) of 5) and etoposide (100 mg/m^2^ daily on days 1–3) with a combination of an ICI (atezolizumab at a dose of 1200 mg on day 1) was delivered without immediate side effects. Nevertheless, the following day the patient presented consciousness disorders (a Glasgow score of 5/15) associated with a partial epileptic seizure (tonic-clonic seizure of the right upper limb), that was secondarily generalized. The partial epileptic seizure persisted despite optimal management with the administration of clonazepam and levetiracetam. Therefore, the patient was transferred to the intensive care unit (ICU) where he received phenytoin and corticosteroids. His arterial blood pressure was elevated (206/108 mmHg), with increased heart and respiratory rate but without fever. On neurological examination, he demonstrated a right hemiplegia, a facial paralysis of the right hemiface and a right pyramidal syndrome. Glycemia was normal. The electroencephalograms performed on the first and second day of the symptoms are shown in Figure 1.

The initial brain computed tomography (CT), pre- and post-contrast administration, was performed on a SOMATOM Force (Siemens Healthineers, Forchheim, Germany) CT scanner at 12 mAs and 3 kV. Images were reconstructed with filtered back projection (FBP) and the advanced modeled iterative reconstruction (ADMIRE; Siemens Healthineers). It showed no evidence of stroke, bleeding or brain metastasis. The Circle of Willis was permeable with the detection of an atheromatous infiltration of the carotid bulbs with a loose stenosis. The electroencephalogram showed a left occipital status epilepticus.

Brain MRI (Magnetic Resonance Imaging) slides are shown in Figure 2, Figure 3 and Figure 4. MR acquisitions were performed on an imaging machine (MRI) from General Electric, Milwaukee, WI, USA: Discovery MR 750w 3T. MRI data included a post-contrast (gadoterate meglumine, Dotarem, Guerbet, Villepinte, France) three-dimensional T1-weighted fast spoiled gradient recalled (FSPGR) acquisition (post-contrast 3DT1), post-contrast 3DT1, and fat-suppressed fluid attenuated inversion recovery (FLAIR) images. To ensure image quality, neuro-radiologists analyzed all the available imaging sequences. Table 1 details the MRI parameters machine. There were multiple bilateral subcortical, parietal, temporal, occipital and cerebellar T2 FLAIR high signals, predominantly in the posterior region with a slight right occipital cortex involvement but without translation in diffusion sequences. There were no abnormal T1 signals nor bleeding, but a left thalamic apparent diffusion coefficient (ADC) restriction was noted. On arterial spin labelling (ASL) perfusion sequences, there was an increased perfusion within the left temporo-parietal-occipital, left thalamic and right cerebellar regions; there was no thrombus on the TOF (time of flight) sequence. Laboratory evaluation for autoimmune, infectious and vascular secondary hypertension etiologies were found to be within normal limits.

In the ICU, the blood pressure was optimally controlled and the antiepileptic therapy was adapted, thus leading to the progressive resolution of the neurological symptoms. Intravenous steroids were stopped and intravenous anti-hypertensive therapy was replaced by oral therapy. The control MRI, performed on the 5th and 12th day with the same MRI parameters, demonstrated the progressive disappearance of the bilateral subcortical posterior T2 high signals and the left thalamic ADC restriction (Figure 5).

The clinical context (high blood pressure, seizure) and radiological features, as well as the progressive resolution of the neurological symptoms, were all in favor of posterior reversible encephalopathy syndrome (PRES). The brain MRI findings with the T2 subcortical posterior high signals, without ADC restriction, reflected the presence of vasogenic edema. The symmetrical posterior distribution, sparing the calcarine and paramedian occipital lobe, without ADC restriction in front of the FLAIR signals, as well as the reversible nature of these abnormalities were not in favor of any of the differential diagnosis of PRES (posterior circulation infarct, progressive multifocal leukoencephalopathy, gliomatosis cerebri, severe hypoglycemia or inflammatory cerebral amyloid angiopathy). The clinical context and the absence of fever did not suggest the diagnosis of herpetic encephalopathy. 

Finally, the neurological symptoms completely regressed after several days of optimal antihypertensive and antiepileptic treatment, thus confirming the diagnosis of PRES of undetermined origin that was favored by the hypertensive terrain, the intake of corticosteroids and the administration of anti-cancer drugs. The multi-disciplinary meeting specialized in immunotherapy-related toxicity did not find any sufficient data to incriminate atezolizumab as the responsible agent for the occurrence of PRES in this patient. Therefore, chemotherapy combined with atezolizumab was then reintroduced with close monitoring of blood pressure without any significant complications.

## 3. Discussion

Posterior reversible encephalopathy syndrome (PRES) was first described in 1996 among 15 patients that had received immunosuppressive therapy, a diagnosis of eclampsia or an acute hypertension associated with renal disease [1]. Several neurological symptoms can occur with encephalopathy, seizure, headache, altered mental function and visual disturbances, being the most frequent [2,3,4]. These symptoms are classically associated with hypertension and renal failure. Nevertheless, the pathophysiology of PRES is not yet well elucidated. An acute hypertensive peak would lead to the disruption of the blood– brain barrier secondary to the inability of the posterior circulation to auto-regulate its blood flow, which would result in vasogenic edema (Figure 6). The posterior blood circulation would be the most affected due to the weak sympathetic innervation [12]. Endothelial dysfunction is also involved in the development of PRES, regardless of blood pressure.

Brain imaging, particularly MRI, is the optimal radiological imaging to support the diagnosis of PRES and eliminate other differential diagnosis [13,14,15,16,17]. The vasogenic edema in PRES is characterized by bilateral hyper-intensity on FLAIR images in the parietal and occipital subcortical white matter [18] but low or iso-intense signals on T1-weighted MRI images, usually without ADC restriction. The calcarine and paramedian part of the occipital lobe is typically spared, which could be useful to differentiate from a bilateral posterior cerebral artery territory infarction [19]. The cerebellum and brainstem may be involved, and more occasionally, the frontal and temporal lobes in the most severe cases [4,13]. Involvement may sometimes be limited to infratentorial structures [20]. The grey matter can also be involved [3,13]. The reversible nature of these MRI abnormalities is highly suggestive of PRES. However, ADC restriction can be observed, suggesting ischemia or cytotoxic edema, most often associated with irreversible damage [12,13,18]. Reversible cerebral vasoconstriction syndrome (RCVS) is the main differential diagnosis with common clinical and radiological features but epileptic seizures and visual disturbances are uncommon and the distribution of lesions is usually asymmetric [21]. RCVS is characterized by vessel irregularities with vasoconstriction or “string-of-beads” appearance which can be visualized by angiography (catheter, magnetic resonance or computed tomography angiography). Several algorithms have been validated to guide clinicians to the optimal diagnosis of PRES [12,18]. Severe hypertension, renal failure, eclampsia, autoimmune disorder, transplantation, infection, and immunosuppressant therapy or cytotoxic drugs are common PRES risk factors [3].

As for the pathophysiology of PRES, the increase of the cerebral blood flow (CBF) on the ASL sequences in the left temporo-parietal-occipital and the left thalamic reflected the increased cerebral perfusion in response to the excessive metabolic demands secondary to the seizure activity. These changes can lead to vasogenic and/or cytotoxic edema. Furthermore, the left thalamic increased DWI signal and the reduced ADC value in the pulvinar region was related to the status epilepticus. Indeed, the thalamus is involved in the transfer of information between the cortical and subcortical structures of critical activities, and potentially in their regulation and propagation [22]. The contralateral cerebellar increased CBF was due to the crossed cortico-thalamo-rubro-dentato-cerebellar fascicles, also called the crossed cerebellar diaschisis, represented in Figure 7 [23,24].

In oncology patients, chemotherapeutic agents are known to be an etiological factor of PRES, such as cytarabine, cisplatin, gemcitabine, and bevacizumab [25]. Several case reports of post-carboplatin or etoposide PRES are published in the literature [5,6,7,8,9,26,27]. In addition, PRES was reported in the combination of carboplatin with a taxane in two patients [5,6] and gemcitabine in one patient [7]. These previous three cases occurred after several cycles of chemotherapy while Ryan et al. reported the case of a patient developing PRES two weeks after the first cycle of carboplatin and etoposide [9]. On the other hand, few cases of PRES occurred after immunotherapeutic agents but autoimmune disorders are commonly reported in patients with PRES [12]. As for immune checkpoint inhibitors, data remains sparse as to the association of immunotherapy and the occurrence of PRES. In fact, two cases of suspected ICI-related PRES were described, the first after the second cycle of ICI combination in a Phase I clinical trial [10] and the second after four cycles of Nivolumab in non-small lung cancer [11].

Finally, there is no specific therapeutic algorithm for PRES. Optimal control of high blood pressure with an adapted antiepileptic management are recommended and, if correctly identified, the triggering factor should be interrupted, at least temporarily, such as chemotherapy in this case. The symptoms usually resolve within days and imaging abnormalities regress more slowly, within days–weeks [3]. It does not appear necessary to prolong anti-epileptic treatment for more than 3 months [3].

## 4. Conclusions

PRES is a rare neurological complication that can occur after the administration of chemotherapy, most commonly with platinum compounds, and/or immunotherapy. Until now, the association of ICIs with PRES had not been validated. Prompt diagnosis is crucial through a spectrum of suspicious clinical and radiological characteristics that must be rapidly recognized to quickly anticipate the optimal therapeutic strategy and avoid unnecessary complications. Therefore, reporting of rare adverse events secondary to ICIs is mandatory for a better understanding of the various clinical spectrums.

## Figures and Tables

**Figure 1 diagnostics-12-01369-f001:**
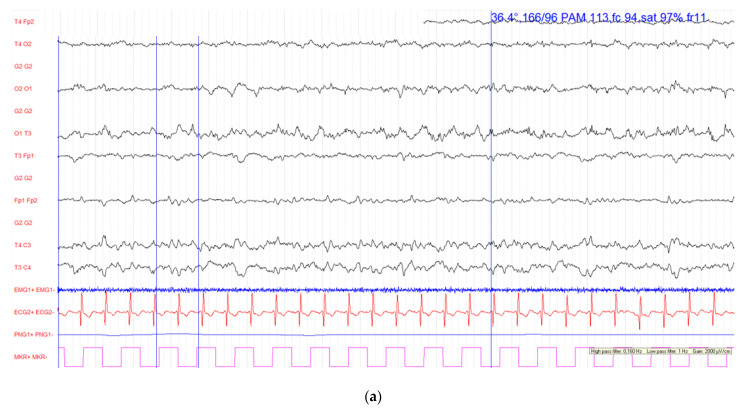
Electroencephalograms (EEG) performed on the first and the second day of symptoms. All channels from R1020 montage are present. EEG configurations are: high pass filter = 0.53 Hz; low pass filter = 70 Hz; scaling = 100 μV/cm. (**a**) First day: Lesional left occipital status epilepticus (**b**) Second day: the background activity appears asymmetrical with a slowed rhythm in the left hemisphere. There are theta delta activities of sometimes periodic expression focused on the left centro-parietal region. Absence of seizure.

**Figure 2 diagnostics-12-01369-f002:**
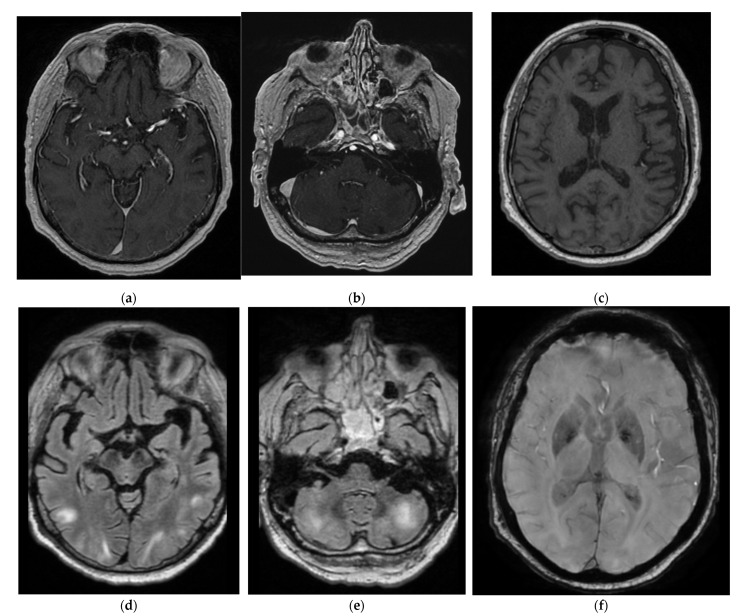
Initial brain magnetic resonance imaging (MRI) with post-contrast T1, T1 SGPR, T2 fluid attenuated inversion recovery (FLAIR) and T2 SWAN sequences: (**a**) Sus-tentorial post-contrast T1 (**b**) Infra-tentorial post-contrast T1 (**c**) Sus-tentorial T1 SPGR (**d**) Sus-tentorial T2 FLAIR (**e**) Infra-tentorial T2 FLAIR (**f**) Sus-tentorial T2 SWAN. We noted multiple bilateral subcortical, parietal, temporal, occipital and cerebellar T2 FLAIR high signals, predominantly in the posterior region. There were no abnormal T1 signals nor bleeding or contrast enhancement.

**Figure 3 diagnostics-12-01369-f003:**
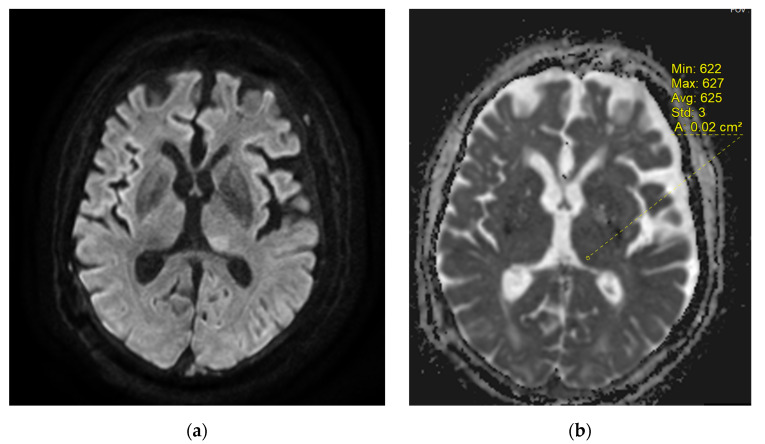
Initial brain magnetic resonance imaging (MRI). (**a**) Diffusion B1000 (**b**) Apparent diffusion coefficient (ADC). We noticed a left thalamic diffusion high signal with ADC restriction.

**Figure 4 diagnostics-12-01369-f004:**
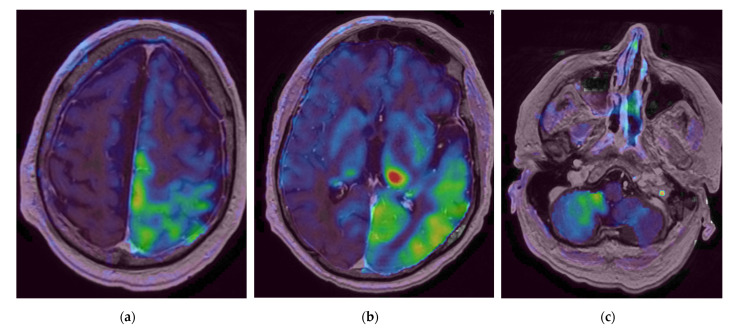
Initial brain Magnetic Resonance Imaging (MRI) with arterial spin labelling (ASL) sequences. (**a**,**b**) Supratentorial acquisitions, (**c**) Infratentorial acquisition. We observe an increased perfusion (represented from green to red) within the left temporo-parietal-occipital, left thalamic and right cerebellar regions.

**Figure 5 diagnostics-12-01369-f005:**
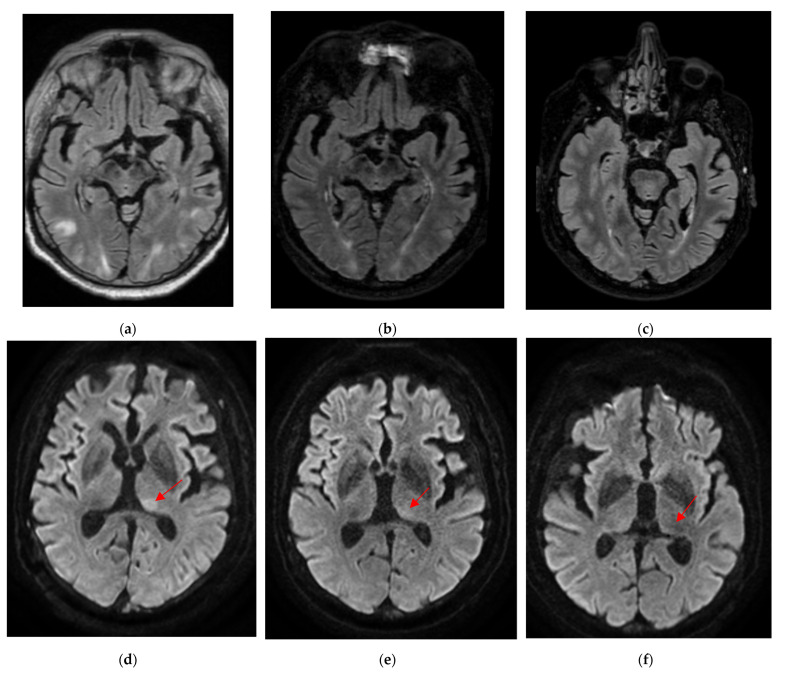
Evolution of brain imaging over time (2nd, 5th and 12th day of onset of symptoms) (**a**) 2nd day T2 fluid attenuated inversion recovery (FLAIR) (**b**) 5th day T2 FLAIR (**c**) 12th day T2 FLAIR (**d**) 2nd day DIFFUSION B1000 (**e**) 5th day DIFFUSION B1000 (**f**) 12th day DIFFUSION B1000. We observed the progressive disappearance of the bilateral subcortical posterior T2 high signals and the left thalamic diffusion high signal (represented by the red arrow).

**Figure 6 diagnostics-12-01369-f006:**
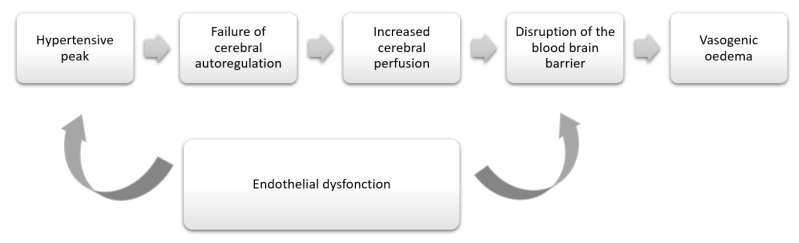
Schematic representation of the pathophysiological mechanisms of posterior reversible encephalopathy syndrome.

**Figure 7 diagnostics-12-01369-f007:**
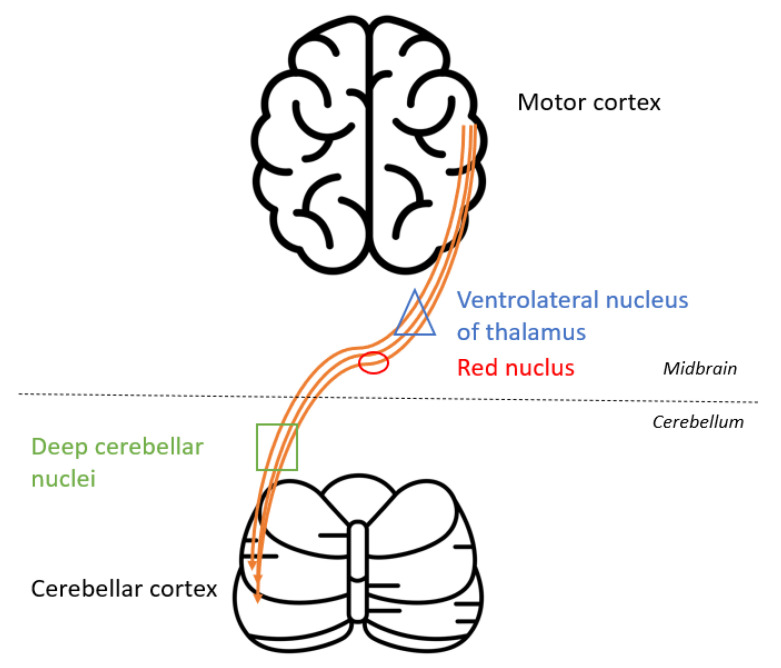
Schematic representation of the crossed cortico-thalamo-rubro-dentato-cerebellar fascicles. Figure 7 has been designed using resources from Flaticon.com (accessed on 18 July 2021).

**Table 1 diagnostics-12-01369-t001:** Magnetic resonance imaging parameters machines. FLAIR: fluid attenuated inversion recovery; DWI: diffusion-weighted imaging; TR: repetition time; TE: echo time; EPI: echo-planar imaging.

Machine	Weighting	Sequence	TR	TE	Slice Thickness
Discovery MR 750w 3T Installed in 2012, 70 cm tunnel, 32 channels, 50 cm z-axisFOV, gradient 44 mT/m SR 200 T/m/s	T1 pre-contrast	3D rapid gradient echo	9 ms	2.1 ms	1 mm
	T2-FLAIR	Turbo spin echo	7002 ms	118 ms	1 mm
	DWI	EPI, two-b-values (0 and 1000 mm/s)	3349 ms	62.6 ms	3 mm
	T1 post-contrast	3D rapid gradient echo	6.1 ms	2.1 ms	1 mm

## Data Availability

Additional data are available on request from the corresponding author.

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
