# Peer review of "Posterior Reversible Encephalopathy Syndrome Following Chemotherapy and Immune Checkpoint Inhibitor Combination in a Patient with Small-Cell Lung Cancer"

_diagnostics, 2022, doi:10.3390/diagnostics12061369_

Round 1
Reviewer 1 Report
General comment:
This case report present a rare neurological complication of posterior reversible encephalopathy syndrome, following bood pressure increase, due to chemotherapy and carboplatine-etoposide-atezolizumab for small cell lung cancer. MRI is used as investigation methodology.
The paper is generally well written and the reading is smooth and fluent.
The referencing must be improved, and the comparison with the state of the art has to be better performed.
Specific comments throughout the paper:
Abstract
Check lines 18-20: probably the size of the font is wrong. Please follow the template.
1. Introduction
Lines 32-33: Missing reference for the symptomatology. Please provide at least a work.
The authors clearly pointed out that the case under analysis is that of a rare pathology, however, in the Introduction, they cited [2]-[8] withouth addressing the problem of a proper comparison with the state-of-the-art, discussing factors such as age, ICIs, investigation methodology. I suggest to provide a summary table for a complete and exhaustive comparison.
2. Case Report
Line 61: Extra space. Please check the editing.
Line 62: Missing details for the CT exam. Please provide all the specifications for the protocol followed for the imaging.
Line 65: Please provide the EEG signals records, as well as the type of electrodes and the machinery used for the biosignal acquisition.
Line 66: Please modify in "Fig. 1-3".
Lines 66-75: Missing details for the MRI apparatus, the RF coils, the quantitative parameter sequences, the post-processing procedures and the software used. I suggest to provide a summary table, it would be bettere for the readers interested in the technical aspects. Please, consider this point to be mandatory since the reproducibility and methodological clarity is fundamental from a scientific point of view.
Fig. 2, pg. 3: Check the font type and size. Please revise.
In Fig. 3 is missing the colorbar. Please add it for the sake of clarity and completness.
Line 91: please specify if the same identical MRI apparatus, RF coils, sequences and software were used.
A quantitative analysis on the MRI images was not performed. This lowers the overall quality and added value.
3. Discussion
Line 131: This bold sentence must be supported with the right references. To me, they are missing. Some counter-examples must be reported. The authors are citing work which used MRI. A deep and coherent discussion must be provided about this point.
4. Conclusion
The conclusion are fine.
I suggest to include some future perspectives to slighlty improve this section.
Author Response
The authors would like to thank the editor and reviewers for their insightful and constructive comments. The authors hope that they have been able to satisfactorily address all of the reviewers’ concerns.
Reviewer 1
This case report present a rare neurological complication of posterior reversible encephalopathy syndrome, following bood pressure increase, due to chemotherapy and carboplatin-etoposide-atezolizumab for small cell lung cancer. MRI is used as investigation methodology.
The paper is generally well written and the reading is smooth and fluent.
The referencing must be improved, and the comparison with the state of the art has to be better performed.
Specific comments throughout the paper:
Abstract
Check lines 18-20: probably the size of the font is wrong. Please follow the template. Corrected
- Introduction
Lines 32-33: Missing reference for the symptomatology. Please provide at least a work. Corrected
The authors clearly pointed out that the case under analysis is that of a rare pathology, however, in the Introduction, they cited [2]-[8] without addressing the problem of a proper comparison with the state-of-the-art, discussing factors such as age, ICIs, investigation methodology. I suggest to provide a summary table for a complete and exhaustive comparison. We agree that adding a table would be helpful. But in our case, we do not discuss a novel therapeutic or diagnostic approach for PRES but the novel occurrence of a rare syndrome with immunotherapy. Thank you for your comment.
- Case Report
Line 61: Extra space. Please check the editing. Corrected
Line 62: Missing details for the CT exam. Please provide all the specifications for the protocol followed for the imaging. Corrected.
Line 65: Please provide the EEG signals records, as well as the type of electrodes and the machinery used for the biosignal acquisition. We agree that this would have been very interesting but unfortunately we were not able to get this information in the time available.
Line 66: Please modify in "Fig. 1-3". Corrected
Lines 66-75: Missing details for the MRI apparatus, the RF coils, the quantitative parameter sequences, the post-processing procedures and the software used. I suggest to provide a summary table, it would be better for the readers interested in the technical aspects. Please, consider this point to be mandatory since the reproducibility and methodological clarity is fundamental from a scientific point of view. Corrected
Fig. 2, pg. 3: Check the font type and size. Please revise. Corrected
In Fig. 3 is missing the colorbar. Please add it for the sake of clarity and completeness. We have completed the legend for clarity.
Line 91: please specify if the same identical MRI apparatus, RF coils, sequences and software were used. Corrected
A quantitative analysis on the MRI images was not performed. This lowers the overall quality and added value.
- Discussion
Line 131: This bold sentence must be supported with the right references. To me, they are missing. Some counter-examples must be reported. The authors are citing work which used MRI. A deep and coherent discussion must be provided about this point. Thank you for your suggestion. We have implemented the bibliography and completed this paragraph.
- Conclusion
The conclusion are fine. I suggest to include some future perspectives to slightly improve this section.
Reviewer 2 Report
The manuscript by Cécile Evin et al., entitled “Posterior Reversible Encephalopathy Syndrome following chemotherapy and immune checkpoint inhibitor combination in a patient with small-cell lung cancer” is the report of a case of posterior reversible encephalopathy syndrome. The case shows a rare pathology chronologically related to the combination of chemotherapy + ICIs. The manuscript is well written, the figures are well presented and signify.
Author Response
The authors would like to thank the editor and reviewers for their insightful and constructive comments. The authors hope that they have been able to satisfactorily address all of the reviewers’ concerns.
The manuscript by Cécile Evin et al., entitled “Posterior Reversible Encephalopathy Syndrome following chemotherapy and immune checkpoint inhibitor combination in a patient with small-cell lung cancer” is the report of a case of posterior reversible encephalopathy syndrome. The case shows a rare pathology chronologically related to the combination of chemotherapy + ICIs. The manuscript is well written, the figures are well presented and signify.
We would like to sincerely thank you for your comments and remarks. We hope that you will be satisfied with our implementations.
Round 2
Reviewer 1 Report
This aspect still need to be addressed:
Comment: Line 65: Please provide the EEG signals records, as well as the type of electrodes and the machinery used for the biosignal acquisition.
Reply: We agree that this would have been very interesting but unfortunately we were not able to get this information in the time available
Author Response
Comment: Line 65: Please provide the EEG signals records, as well as the type of electrodes and the machinery used for the biosignal acquisition.
Thank you for giving us the extra time. We have added the electroencephalograms with the parameters used and their analysis.
Round 3
Reviewer 1 Report
I thank the authors for providing the missing information.
No further comments from me.
This manuscript is a resubmission of an earlier submission. The following is a list of the peer review reports and author responses from that submission.
Round 1
Reviewer 1 Report
PRESS in the setting of chemotherapy and immuno-therapy is uncommon but not that rare. In the current day and age, this occurrence had become rather prosaic. I do not believe that this manuscript adds significantly to the existing scientific literature.
Reviewer 2 Report
The manuscript under review is the report of a case of posterior reversible encephalopathy syndrome. The case shows a rare pathology chronologically related with the combination of chemotherapy + ICIs. I think the case would not attract the attention of many readers since it is on a very specific and rare pathology. However, the manuscript is well-written, the figures are well presented and significative. I have a few minor comments to improve the manuscript:
- I think the case report should not divided into paragraphs with subtitles. I suggest unifying all the paragraph 2.1…2.2 … etc deleting the subtitles.
- I think a figure that shows a schematic representation of the pathophysiological mechanisms of PRES would be interesting and would add value to the manuscript.
- I think the reference list should be implemented. I suggest the following papers: 1007/s00415-016-8377-8; 10.1111/bpa.13013; 10.1111/j.1552-6569.2004.tb00223.x; 10.14670/HH-18-319; 10.1136/practneurol-2011-000010